# Rit1-TBC1D10B signaling modulates FcγR-mediated phagosome formation in RAW264 macrophages

Youhei Egami◉, Katsuhisa Kawai, Nobukazu Araki

**Phagocytosis is an important immune response that protects the host from pathogen invasion. Rit1 GTPase is known to be involved in diverse cellular processes. However, its role in FcγR-mediated phagocytosis remains unclear. Our live-cell imaging analysis revealed that Rit1 was localized to the membranes of F-actin-rich phagocytic cups in RAW264 macrophages. Rit1 knockout and expression of the GDP-locked Rit1 mutant suppressed phagosome formation. We also found that TBC1D10B, a GAP for the Rab family GTPases, colocalizes with Rit1 in the membranes of phagocytic cups. Expression and knockout studies have shown that TBC1D10B decreases phagosome formation in both Rab-GAP activity–dependent and –independent manners. Notably, the expression of the GDP-locked Rit1 mutant or Rit1 knockout inhibited the dissociation of TBC1D10B from phagocytic cups. In addition, the expression of the GTP-locked Rit1 mutant promoted the dissociation of TBC1D10B in phagocytic cups and restored the rate of phagosome formation in TBC1D10B-expressing cells. These data suggest that Rit1-TBC1D10B signaling regulates FcγR-mediated phagosome formation in macrophages.**

## Introduction

Phagocytosis is a specialized endocytic pathway that allows the ingestion of large particles (>1 $\mu$m in diameter) and plays a critical role in innate immunity and tissue remodeling. Professional phagocytes, mainly macrophages and neutrophils, have the ability to phagocytose invading microorganisms, foreign particles, and apoptotic bodies, contributing to the resolution of infections and clearance of damaged or senescent cells. Phagocytosis is induced by the binding of a particle to a specific cell surface receptor, such as Fcγ receptors, complement receptors (CRs), scavenger receptors, and Toll-like receptors (Greenberg & Grinstein, 2002; Freeman & Grinstein, 2014; Lim et al, 2017). Phagocytosis through FcγRs is the best-characterized endocytic pathway, which is accompanied by actin polymerization and reorganization to form phagosomes (Campellone & Welch, 2010). After the formation of F-actin-rich phagocytic cups extending along opsonized particles and the subsequent closure of phagocytic cups into internalized phagosomes, which are not surrounded by F-actin, the nascent phagosomes gradually mature by a series of interactions with intracellular endocytic compartments, eventually, fuse with lysosomes, and lead to particle degradation (Pauwels et al, 2017).

Ras family GTPases mainly function as molecular switches that alternate between inactive GDP- and active GTP-bound forms. Guanine nucleotide exchange factors and GTPase-activating proteins (GAPs) regulate the switching between these states. The active GTP-bound forms directly interact with their downstream effectors, thereby activating specific signaling pathways. Within the Ras family, Rit1 and Rit2 share many features. Rit1 is widely expressed in various tissues (Wes et al, 1996). In contrast, Rit2 is specifically expressed in neurons (Lee et al, 1996). It has been shown that Rit1 is involved in cell survival in response to stress, neuronal morphogenesis, and neural differentiation (Spencer et al, 2002; Shi et al, 2005, 2011, 2012; Cai & Andres, 2014). Notably, a previous study indicated that Rit1 regulates actin cytoskeletal rearrangement via the p21-activated protein kinase 1 (PAK1)/Rac1/Cdc42 complex and promotes cell motility (Meyer Zum Büschenfelde et al, 2018). These findings imply that Rit1 has various cellular and physiological functions. However, the immunological function of Rit1 has not yet been addressed in macrophages.

Tre-2/Bub2/Cdc16 (TBC) domain-containing proteins function as GAPs for Ras-related Rab GTPases (Fukuda, 2011). More than 40 TBC domain-containing members have been identified in mammals and are involved in diverse membrane trafficking events (Fukuda, 2011). Among them, TBC1D10 subfamily members (TBC1D10A/EPI64, TBC1D10B/EPI64B, and TBC1D10C/EPI64C/carabin), which have a TBC domain-encoding catalytic activity, show GAP activity toward Rab and Ras GTPases. TBC1D10A regulates melanosome transport via the inactivation of Rab27A (Itoh & Fukuda, 2006), exosome secretion, selective autophagy mediated by Rab35 inactivation (Hsu et al, 2010; Minowa-Nozawa et al, 2017), and VEGFR2 trafficking by acting on Rab13 (Xie et al, 2019). TBC1D10B has been implicated in modulating intracellular membrane transport by Rab3A, Rab22A, Rab27A, Rab27B, and Rab35 in various cell types (Ishibashi et al, 2009; Hsu et al, 2010; Hou et al, 2013). TBC1D10C controls receptor recycling and immunological synapse formation in T cells

Department of Histology and Cell Biology, School of Medicine, Kagawa University, Miki, Japan

Correspondence: egami.yohei@kagawa-u.ac.jp

(Patino-Lopez et al, 2008). Notably, TBC1D10C possesses Ras GAP activity and inhibits the Ras signaling pathway on T-cell activation (Pan et al, 2007). Furthermore, Nagai et al have reported that TBC1D10A and TBC1D10B also exhibit GAP activity toward several Ras GTPases (H-Ras, K-Ras, and N-Ras) (Nagai et al, 2013). In macrophages, TBC1D10C has been shown to participate in the phagocytosis of *Burkholderia cenocepacia* (Villagomez et al, 2021). However, the roles of TBC1D10 subfamily members (TBC1D10A, TBC1D10B, and TBC1D10C) in FcγR-mediated phagocytosis have not been investigated. In this study, we provide novel evidence that Rit1-TBC1D10B signaling regulates phagosome formation in macrophages during FcγR-mediated phagocytosis.

## Results

### Rit1 is localized to the membrane of F-actin–rich phagocytic cups during FcγR-mediated phagocytosis in macrophages

It has been shown that Rit1 is widely expressed in most tissues (Wes et al, 1996). However, the expression of Rit1 in macrophages remains unknown. We first examined the expression of Rit1 in bone marrow–derived macrophages (BMMs) and RAW264 macrophages. As shown in Fig 1A and B, Rit1 was detected in BMMs and RAW264 macrophages by RT–PCR and Western blot analysis. The expression of Rit2 was undetectable in these cells (data not shown). Next, we investigated whether Rit1 is involved in FcγR-mediated phagocytosis. RAW264 cells were transfected with GFP-Rit1 WT and observed using confocal laser microscopy. Before phagocytosis, GFP-Rit1 was mainly localized in the plasma membrane and cytosol. After the addition of IgG erythrocytes (IgG-Es) to the cells, Rit1 was recruited to the membranes of phagocytic cups (Fig 2, t = 3 min; Video 1). Subsequently, Rit1 dissociated from the membranes of the nascent and internalized phagosomes (Fig 2, t = 6 min). Phagocytosis via FcγRs is accompanied by actin polymerization and reorganization, which induces the formation of phagocytic cups around IgG-opsonized targets (Araki et al, 1996, 2003). Recently, it has been shown that Rit1 is involved in actin cytoskeletal rearrangement during cell migration (Meyer Zum Büschenfelde et al, 2018). Thus, we compared the subcellular localization of Rit1 and F-actin during phagosome formation. Time-lapse observation of live RAW264 macrophages co-expressing TagRFP-Rit1 and GFP-LifeAct, which labels F-actin, showed that Rit1 colocalized with F-actin in the phagocytic cup (Fig 3, t = 3 min; Video 2). Thereafter, both proteins dissociated from the newly formed phagosomes after closing the cups to phagosomes (Fig 3, t = 8 min).

### Expression of GDP-bound mutant Rit1-S35N or endogenous Rit1 knockout inhibits FcγR-mediated phagocytosis

The localization of Rit1 in the early phase of IgG-E uptake by RAW264 cells suggests that Rit1 may be involved in phagosome formation. To clarify the functional relevance of Rit1 in phagosome formation, we examined the effects of Rit1 mutant expression on IgG-E internalization into phagosomes. We transiently transfected RAW264 cells with Rit1-WT, GTP-bound mutant Rit1-Q79L, or GDP-bound

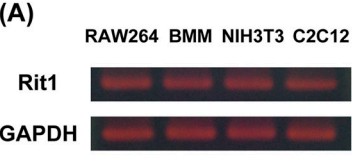

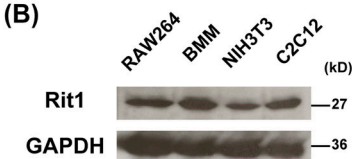

**Figure 1. Rit1 is expressed in RAW264 cells and bone marrow–derived macrophages.**

**(A)** RT–PCR analysis of Rit1 mRNA expression in RAW264 cells, bone marrow–derived macrophages, NIH3T3 cells, and C2C12 myoblasts was conducted. GAPDH mRNA was used as an internal control. Similar results were obtained from three independent experiments. **(B)** Comparison of Rit1 protein levels in RAW264 cells, bone marrow–derived macrophages, NIH3T3 cells, and C2C12 myoblasts. Rit1 protein was detected by Western blotting using an anti-Rit1 antibody. GAPDH was used as an internal control for normalization. Notably, Rit1 is expressed in RAW264 cells and bone marrow–derived macrophages.

mutant Rit1-S35N. Then, a quantitative assay of phagosome formation and IgG-E binding was conducted in cells expressing Rit1 in each construct. As shown in Fig 4 (gray bars), the expression of the GDP-bound mutant Rit-S35N inhibited IgG-E uptake. In contrast, the expression of neither Rit1-WT nor the GTP-bound mutant Rit1-Q79L had an effect on phagosome formation. Binding of IgG-Es to cells was not significantly affected by the expression of Rit1-WT, Rit1-Q79L, or Rit-S35N (Fig 4, open bars). Using the CRISPR/Cas9 system, we next generated Rit1-KO-RAW264 macrophages to substantiate the significance of endogenous Rit1 GTPase in phagosome formation. As shown in Fig 5A, three KO clones did not show any detectable Rit1 protein, as assessed by Western blot analysis. Quantification of phagocytosis revealed that the engulfment of IgG-Es was inhibited in Rit1-deficient macrophages (Fig 5B). The knockout of Rit1 had no effect on the binding of IgG-E to the cells. To further exclude the possibility that phagocytosis was suppressed by undesired off-target effects, rescue experiments were performed using Rit1-KO cells. Fig 5C shows that transient expression of Rit1-WT in Rit1-deficient cells restored FcγR-mediated phagosome formation. We also found that the expression of the GTP-bound mutant Rit1-Q79L, and Rit1-WT, recovered phagosome formation in Rit1-KO cells, whereas Rit1-WT-W204A or Rit1-Q79L-W204A, which showed a loss of plasma membrane (PM) targeting (Heo et al, 2006), did not restore phagocytosis in the Rit1-deficient cells (Fig S1). These results indicate that Rit1 is an important constituent of FcγR-mediated phagocytosis and controls phagosome formation.

### TBC1D10B is transiently recruited to the phagosomal membrane during FcγR-mediated phagocytosis

Previous reports have shown that TBC1D10A, TBC1D10B, and TBC1D10C (TBC1D10A/B/C) exhibit GAP activity toward several Ras

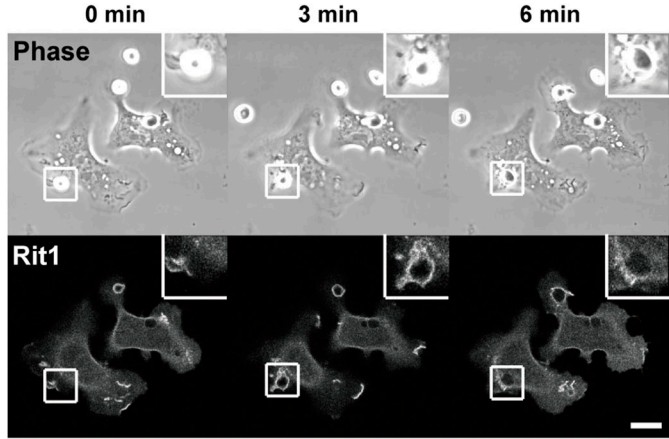

**Figure 2. GFP-Rit1 is localized to membranes that form phagosomes during FcγR-mediated phagocytosis.**
RAW264 cells transfected with GFP-Rit1 were exposed to IgG-Es and observed by laser confocal microscopy. The upper panels show the phase-contrast images. The elapsed time is indicated at the top of the figure. The binding of IgG-Es to the cell surface, marked by squares, was set as time 0. It is noteworthy that Rit1 is associated with membranes that form phagosomes (t = 3 min). Between 0 and 30 phagosomes per cell were observed. The insets show higher-magnification images of the boxed regions of the cells. The upper and lower panels show representative images from four independent experiments. The corresponding video is shown in Video 1. Scale bar: 10 μm.

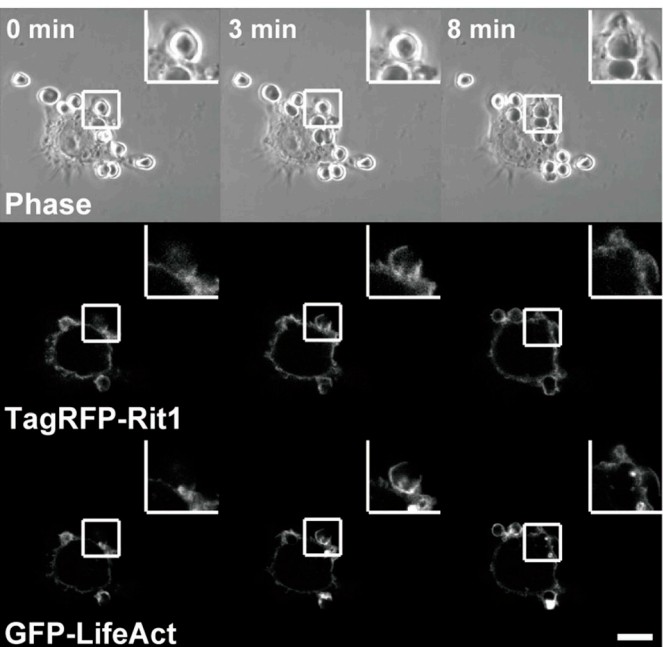

**Figure 3. Rit1 and F-actin colocalize in phagocytic cups.**
Live RAW264 macrophages co-expressing TagRFP-Rit1 and GFP-LifeAct were put into contact with IgG-Es and observed by confocal microscopy. The top panels show the phase-contrast images. The elapsed time is indicated at the top. TagRFP-Rit1 and GFP-LifeAct colocalized at the phagocytic cup (t = 3 min). Subsequently, both proteins were detached from the membranes of the newly formed phagosomes. Between 0 and 30 phagosomes per cell were observed. Images are representative of three independent experiments. The corresponding video is shown in Video 2. Scale bar: 10 μm.

family members (Pan et al, 2007; Nagai et al, 2013). Furthermore, it has been demonstrated that TBC1D10C is involved in *B. cenocepacia* phagocytosis in macrophages (Villagomez et al, 2021). Therefore, we predicted that TBC1D10A/B/C may be a regulator of Rit1 GTPase during phagocytosis. Because the participation of TBC1D10A/B/C in FcγR-mediated phagocytosis is unknown, we first observed the spatiotemporal localization of TBC1D10A/B/C in live RAW264 macrophages expressing each TBC1D10 isoform tagged with GFP during the process of IgG-E uptake. Before the beginning of phagocytosis, TBC1D10A was localized in the cytosol, nucleus, and plasma membrane (Fig 6A). TBC1D10B was mainly found in the cytosol and plasma membrane and was not localized in the nucleus (Fig 6B). TBC1D10C was diffusely localized in the cytosol and nucleus before phagocytosis (Fig 6C). After the binding of IgG-Es to the cell membrane, both TBC1D10A and TBC1D10B accumulated in the membranes of phagocytic cups, extending along the surface of IgG-Es (Fig 6A and B). TBC1D10C was slightly localized in phagocytic cups (Fig 6C). Thereafter, TBC1D10A/B/C readily dissociated from the membranes of newly internalized phagosomes. Quantitative analysis of the fluorescence intensity of GFP-TBC1D10A/B/C revealed that TBC1D10B was robustly recruited to the membrane of the phagocytic cup, in contrast to TBC1D10A/C (Fig 6D).

### Expression of TBC1D10B-WT or GAP activity–deficient TBC1D10B-R395A mutant inhibits FcγR-mediated phagocytosis

Based on the localization of TBC1D10A/B/C at the early stage of FcγR-mediated phagocytosis, we tested the effects of TBC1D10A/B/C WT and their GAP activity–deficient mutants on phagosome formation to ascertain the functional relevance of these TBC1D10

isoforms. We transiently transfected RAW264 cells with TBC1D10A-WT/-R160A, TBC1D10B-WT/-R395A, or TBC1D10C-WT/-R139A. Then, a quantitative assay of IgG-E uptake and IgG-E binding was performed in cells expressing TBC1D10A, TBC1D10B, or TBC1D10C constructs. As shown in Fig 7 (gray bars), the expression of TBC1D10A-WT, TBC1D10B-WT, and TBC1D10C-WT inhibited phagosome formation. Importantly, a decrease in IgG-E uptake was also observed in cells expressing the GAP activity–deficient mutant TBC1D10B-R395A, whereas the expression of neither TBC1D10A-R160A nor TBC1D10C-R139A affected the rate of phagosome formation. In contrast to the effects on the uptake of particles, the expression of TBC1D10A, TBC1D10B, and TBC1D10C did not significantly affect the binding of particles to cells (Fig 7, open bars). As the inhibitory effect of TBC1D10B expression on phagosome formation was more evident than that of TBC1D10A or TBC1D10C expression (Fig 7, gray bars), we examined the physiological significance of TBC1D10B during FcγR-mediated phagocytosis. To demonstrate the importance of endogenous TBC1D10B in phagosome formation, TBC1D10B-KO-RAW264 macrophages were generated using CRISPR/Cas9. As shown in Fig 8A, the absence of TBC1D10B in RAW264 cells was confirmed by Western blot. A quantitative phagocytosis assay revealed that the uptake of IgG-Es was promoted in TBC1D10B-deficient macrophages (Fig 8B). These data indicate that TBC1D10B inhibits FcγR-mediated phagosome formation in both Rab-GAP activity–dependent and –independent manners.

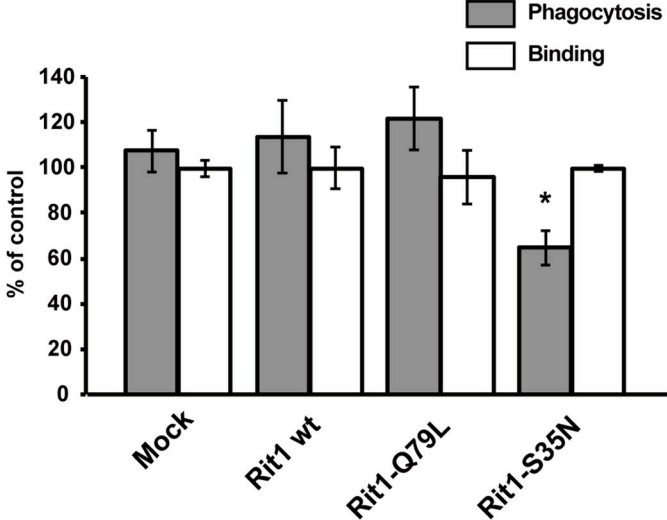

**Figure 4. Expression of the GDP-bound mutant Rit1-S35N inhibits phagosome formation.**
To quantify FcγR-dependent phagosome formation, RAW264 cells transiently expressing GFP (mock), GFP-Rit1-WT, GFP-Rit1-Q79L, or GFP-Rit1-S35N were incubated with IgG-Es for 20 min at 37°C. The cells on the coverslips were fixed after disruption of extracellularly exposed IgG-Es. The efficiency of IgG-E engulfment (gray bars) was calculated based on 50 transfected and 50 untransfected cells. The results are presented as the percentage of control (untransfected) cells. The mean ± SEM of four independent replicates is plotted. Note the inhibition of phagosome formation in cells transfected with the GDP-bound mutant of Rit1. For the binding assay, RAW264 cells expressing the indicated proteins were incubated with IgG-Es for 30 min at 4°C. After brief washing in PBS, the cells on the coverslips were fixed. The efficiency of IgG-E binding to cells (open bars) was calculated. The data are presented as the mean ± SEM of three independent replicates. *$P < 0.05$, compared with GFP-transfected cells (mock) (one-way ANOVA followed by Dunnett's test).

**Rit1 is colocalized with TBC1D10B at the phagosomal membrane and regulates phagosome formation via modulating TBC1D10B localization**

Taking our data on the physiological importance of Rit1 and TBC1D10B during FcγR-mediated phagocytosis into consideration, we predicted that Rit1 might be a possible downstream or upstream target of TBC1D10B in the process of phagosome formation. First, we compared the spatiotemporal localization of Rit1 and TBC1D10B during the uptake of IgG-opsonized particles. Time-lapse observation of cells expressing GFP-Rit1-WT and TagRFP-TBC1D10B-WT showed that before the onset of phagocytosis, both Rit1 and TBC1D10B were localized in the cytosol and plasma membrane. After the binding of IgG-Es to the cell membrane, both proteins colocalized at the membranes of phagocytic cups (Fig 9, arrows, Video 3). Thereafter, they readily dissociated from the membranes of the newly internalized phagosomes (Fig 9, arrowheads). Based on the colocalization data, we examined the functional relationships between Rit1 and TBC1D10B. Because TBC1D10A/B/C has been shown to possess GAP activity toward several Ras family members (Nagai et al, 2013), we examined whether TBC1D1010A/B/C proteins have GAP activity toward Rit1 GTPase. To monitor the activation levels of Rit1, we took advantage of the GTP-dependent interaction between Rit1 and RalGDS–like 3 (RGL3)-Ras–binding domain (Shao

& Andres, 2000). In our GST pull-down assay, the expression of TBC1D10B-WT, but not TBC1D10B-R395A, lowered the activation levels of Rab35 (Fig S2A), indicating that TBC1D10B functions as a GAP for Rab35, as previously shown (Egami et al, 2015). However, GTP-Rit1 levels were unaffected in cells expressing TBC1D10B-WT compared with those in cells expressing TBC1D10B-R395A or in non-transfected cells (Fig S2B). Furthermore, neither TBC1D10A nor TBC1D10C expression affected the activation levels of Rit1, implying that Rit1 is not a major downstream target of TBC1D10A/B/C as a GAP (Fig S2B). In view of our finding that endogenous Rit1-KO or the expression of the GDP-bound form Rit-S35N, and TBC1D10B, inhibits phagosome formation, we next validated the effect of Rit1 down-regulation on the localization of TBC1D10B during phagosome formation. Confocal live-cell imaging demonstrated that the dissociation of TBC1D10B from the membrane of phagocytic cups was inhibited in cells expressing the GDP-bound form of Rit-S35N or Rit1-KO cells (Fig 10A and B). In contrast, TBC1D10B dissociated from phagocytic cups in cells expressing the GTP-bound mutant Rit1-Q79L (Fig 10C). Quantification of the dissociation time of TBC1D10B from the phagosomal membrane at the IgG-E binding sites indicated that the expression of the GTP-bound mutant Rit1-Q79L reduced the accumulation time of TBC1D10B (Fig 10D). Given the importance of Rit1 activation in the dissociation of TBC1D10B from phagosomal membranes, we tested the effect of Rit1-Q79L expression on phagosome formation in TBC1D10B overexpressed cells. As shown in Fig 10E, the expression of Rit1-Q79L partially restored phagosome formation in cells expressing TBC1D10B-WT or TBC1D10B-R395A. In contrast, the expression of Rit1-Q79L-W204A, which leads to the loss of plasma membrane targeting, did not restore phagocytic uptake in TBC1D10B overexpressed cells. Taken together, these results suggest that Rit1 modulates FcγR-mediated phagocytosis by regulating TBC1D10B localization.

## Discussion

In this study, we observed that Rit1 was recruited to the membranes of F-actin-rich phagocytic cups during FcγR-mediated phagocytosis. It is known that dynamic changes in the levels of phosphoinositide species are closely related to remodeling of the actin cytoskeleton during phagocytosis (Levin et al, 2015). For instance, the amount of PI(4,5)P$_2$ is locally increased in the membrane of pseudopodia during pseudopod extension driven by actin polymerization. Moreover, PI(3,4,5)P$_3$ accumulates at the bases of phagocytic cups, where F-actin is disassembled during the closure of phagocytic cups into internalized phagosomes. A previous report showed that the C-terminus of Rit1 possesses a unique polybasic region that binds to PI(4,5)P$_2$ and PI(3,4,5)P$_3$ (Heo et al, 2006). Therefore, it is conceivable that the recruitment and localization of Rit1 to the membranes of phagocytic cups are linked to phosphoinositide levels.

We found that the rate of phagosome formation was significantly inhibited by the expression of a GDP-bound mutant of Rit1 or CRISPR/Cas9-mediated knockout of Rit1. Furthermore, the expression of GTP-bound form Rit1-Q79L and Rit1-WT restored phagosome formation in Rit1-KO cells, whereas the expression of

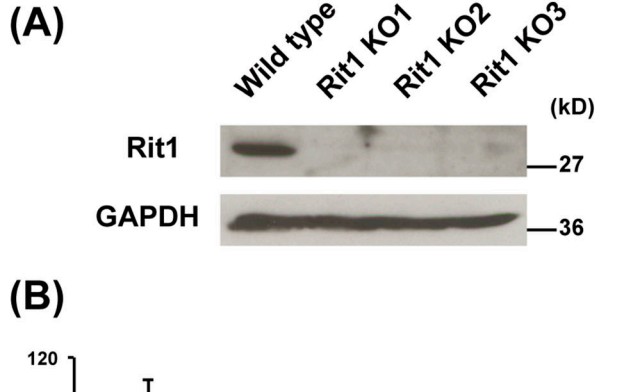

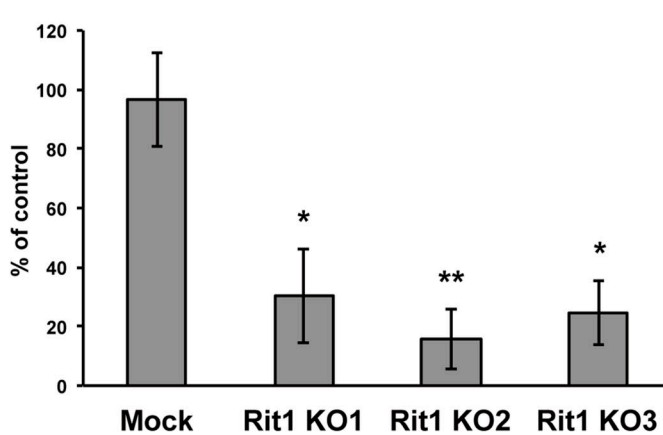

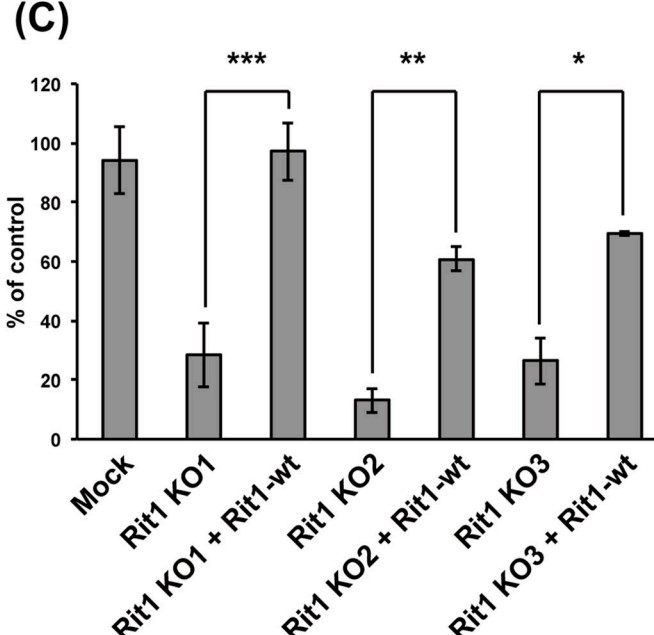

**Figure 5. Depletion of endogenous Rit1 suppresses FcγR-mediated phagocytosis.**

**(A)** RAW264 macrophages transfected with pSpCas9(BB)-2A-puro-Rit1 encoding Cas9 protein and gRNA for Rit1 were diluted and selected as single clones with puromycin. The knockout of the Rit1 protein was validated by Western blotting analysis using an antibody against Rit1. GAPDH was used as the internal control. **(B)** The efficiency of IgG-E uptake was calculated based on 50 cells knocked out for Rit1 and 50 cells transfected with pSpCas9(BB)-2A-puro plasmid (mock) or 50 untransfected (control) cells. The results are displayed as the mean ± SEM percentage compared with control cells for three independent replicates.

Rit1-WT-W204A or Rit1-Q79L-W204A, which leads to loss of plasma membrane localization, did not. These data indicate that Rit1 is an important component of FcγR-mediated phagocytosis and that both activated Rit1 and its localization to the plasma membrane are required for phagosome formation. Because the expression of the GTP-bound form of Rit1-Q79L does not decrease the rate of phagosome formation, it is suggested that the activation–inactivation cycle of Rit1 is dispensable for the uptake of IgG-opsonized particles.

Currently, GAPs and Guanine nucleotide exchange factors specific to Rit1 have yet to be identified. A previous report has demonstrated that TBC1D10 family members TBC1D10A/B/C have GAP activity toward several Ras family members, H-Ras, K-Ras, and N-Ras (Nagai et al, 2013). Our live-cell imaging analysis showed that TBC1D10B was predominantly recruited to the membranes of phagocytic cups. In addition, TBC1D10B and Rit1 colocalized in phagocytic cups during phagosome formation. However, in our pull-down experiments with whole-cell lysates, the expression of TBC1D10B did not promote the inactivation of Rit1 compared with Rab35, implying that Rit1 is unlikely to be a major downstream target of TBC1D10B as a GAP. Based on these findings, we paradoxically considered the possibility that Rit1 functions as an upstream regulator of TBC1D10B. We observed that TBC1D10B predominantly accumulated in the membrane of phagocytic cups and suppressed FcγR-mediated phagocytosis, suggesting that the recruitment of TBC1D10B to the phagosomal membrane is crucial for the inhibition of phagosome formation. Moreover, time-lapse observations showed that the dissociation of TBC1D10B from the phagosomal membrane at the sites of particle binding was inhibited in Rit1-KO cells or cells expressing the GDP-bound mutant Rit1-S35N, whereas the expression of the GTP-bound form Rit1-Q79L facilitated the dissociation of TBC1D10B from the phagosomal membranes during FcγR-mediated phagocytosis. In agreement with these localization data, a quantitative assay of phagosome formation revealed that the expression of the GTP-bound form Rit1-Q79L, but not Rit1-Q79L-W204A, restores phagosome formation in TBC1D10B-expressing cells. Although the precise mechanism regulating the dissociation of TBC1D10B from the phagosomal membrane remains unclear, our experimental data support the notion that Rit1 is the upstream regulator of TBC1D10B and that the localization of TBC1D10B is regulated by Rit1 GTPase in the process of FcγR-mediated phagocytosis.

As mentioned above, we observed that TBC1D10B was transiently recruited to the membranes of phagocytic cups. Interestingly,

*$P < 0.05$; **$P < 0.01$ compared with mock-transfected cells (one-way ANOVA followed by Dunnett's test). **(C)** The efficiency of phagosome formation was calculated based on cells knocked out for Rit1, Rit1-KO cells overexpressing GFP-Rit1-WT, and pSpCas9n(BB)-2A-puro (mock) transfected or untransfected (control) cells. The number of internalized IgG-Es in 50 cells was counted in each replicate. The results are presented as the mean ± SEM of four independent replicates. It is noteworthy that the decrease in phagosome formation in Rit1-KO cells was rescued by overexpression of Rit1-WT. ***$P < 0.001$ compared with Rit1-KO1 cells. **$P < 0.01$ compared with Rit1-KO2 cells. *$P < 0.05$, compared with Rit1-KO3 cells (one-way ANOVA followed by Tukey's test).

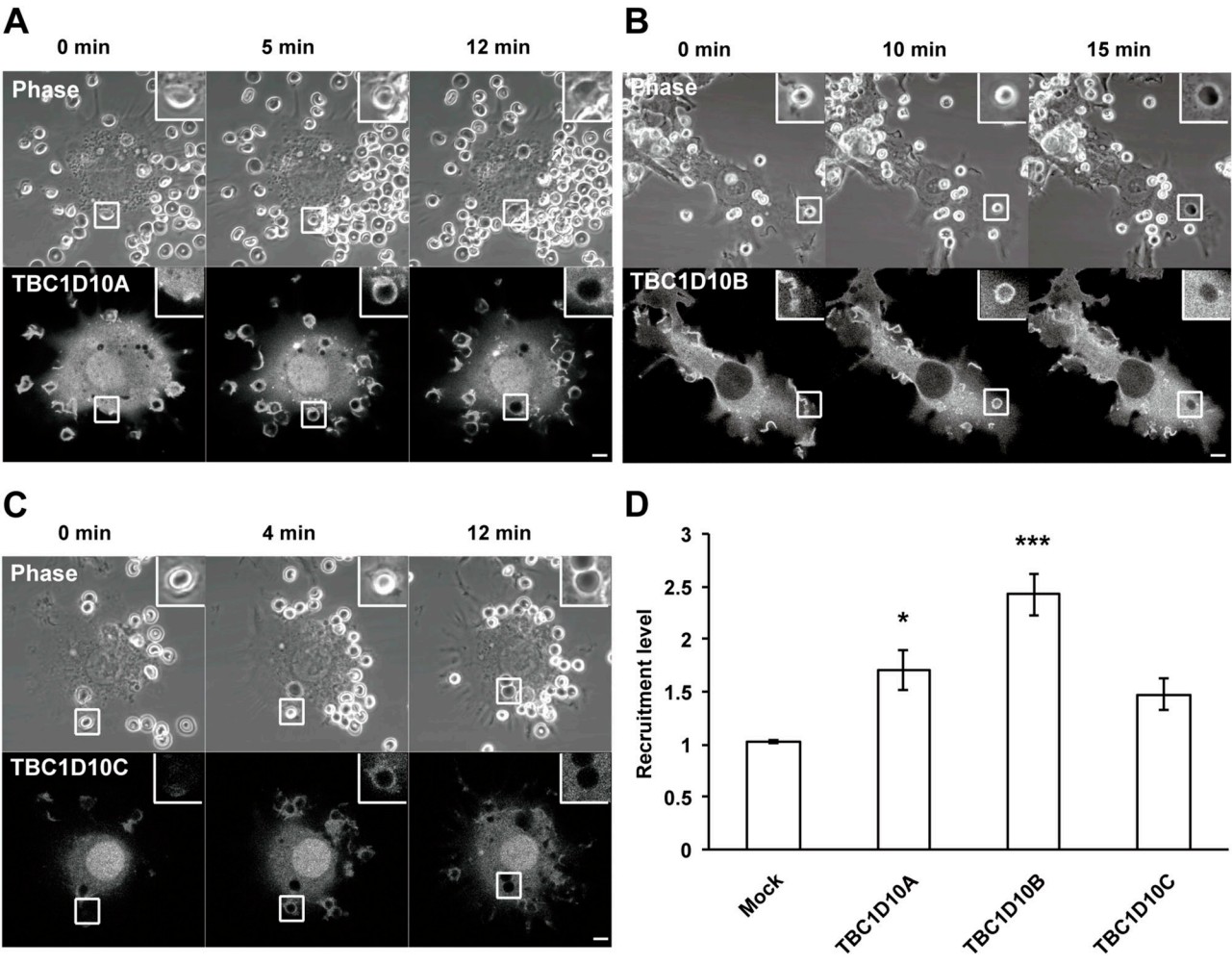

**Figure 6. Localization of TBC1D10A, TBC1D10B, and TBC1D10C during FcγR-mediated phagocytosis.**
**(A, B, C)** Live RAW264 cells transfected with GFP-TBC1D10A (A), GFP-TBC1D10B (B), and GFP-TBC1D10C (C) were allowed to interact with IgG-Es. Time-lapse images of cells expressing each construct indicated were acquired by confocal microscopy. Phase-contrast images are shown in the upper panels. The elapsed time is shown at the top of the figure. Notably, TBC1D10B accumulates intensely in the membranes of forming phagosomes (t = 10 min, (B)). RAW264 cells transfected with GFP-TBC1D10A exhibited between 0 and 6 phagosomes per cell. For cells transfected with GFP-TBC1D10B, between 0 and 3 phagosomes per cell were observed. Cells expressing GFP-TBC1D10C showed between 0 and 8 phagosomes per cell. The upper and lower panels are representative images from four independent experiments. Scale bar: 5 $\mu$m.
**(D)** Quantification of the recruitment levels of TBC1D10A, TBC1D10B, and TBC1D10C at the phagocytic cup. Maximum values of fluorescence intensity were measured at the phagocytic cup in each cell transfected with GFP-TBC1D10A, GFP-TBC1D10B, GFP-TBC1D10C, or GFP (Mock). The fluorescence intensity of GFP-TBC1D10A, GFP-TBC1D10B, GFP-TBC1D10C, and GFP was normalized to that of a region in the cytoplasm. Values represent the mean ± SEM of three independent replicates (n = 3; 30 phagocytic cups in more than five cells in each condition were assessed per replicate). *$P < 0.05$; ***$P < 0.001$ versus corresponding GFP-transfected cells (mock) (one-way ANOVA followed by Dunnett's test).

unlike other TBC1D10 family members TBC1D10A/C, TBC1D10B inhibits phagosome formation in both Rab-GAP activity–dependent and –independent manners. The molecular mechanism by which TBC1D10B regulates phagosome formation has not yet been elucidated. We previously reported that Rab35, one of the downstream targets of TBC1D10 family members, is transiently recruited to the membrane of phagocytic cups and controls FcγR-mediated phagosome formation in macrophages (Egami et al, 2011). Thus, it is possible that TBC1D10B suppresses phagosome formation through inactivation of Rab35 as a substrate. In our study, we found that TBC1D10B expression inhibited phagosome formation independently of its GAP activity. It has been shown that TBC1D10B directly interacts with GTP-bound

form of ARF6 GTPases (Chesneau et al, 2012). Moreover, ARF6 is involved in the regulation of membrane trafficking and actin cytoskeletal remodeling during phagosome formation (Zhang et al, 1998; Niedergang et al, 2003). Therefore, in addition to down-regulating Rab35 activity during FcγR-mediated phagocytosis, TBC1D10B might inhibit phagocytosis by recruiting the activated form of ARF6 to the phagosomal membranes as a scaffold protein.

In conclusion, our study sheds light on the importance of positive and negative regulators Rit1 and TBC1D10B, respectively, during FcγR-mediated phagocytosis. We further provided evidence that Rit1 is localized to the membrane of phagocytic cups and controls FcγR-mediated phagosome formation by

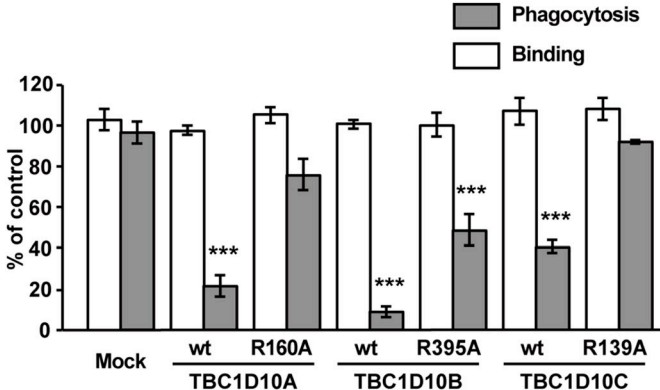

**Figure 7. Effect of TBC1D10A, TBC1D10B, and TBC1D10C expression on FcγR-mediated phagosome formation.**
RAW264 macrophages expressing each construct indicated were incubated with IgG-Es. The efficiency of phagosome formation (gray bars) and IgG-E binding (open bars) was calculated based on 50 transfected cells and 50 non-transfected cells. The results were expressed as a percentage of control (non-transfected) cells. The data shown are the mean ± SEM of three independent experiments. ***$P$ < 0.001 versus corresponding mock-transfected cells (one-way ANOVA followed by Dunnett's test). The expression of TBC1D10A-WT, TBC1D10B-WT, and TBC1D10C-WT has an inhibitory effect on IgG-E uptake. Notably, inhibition of phagosome formation was also found in cells expressing the GAP activity–deficient mutant TBC1D10B-R395A.

facilitating TBC1D10B dissociation from the phagosomal membrane. In future studies, elucidation of the detailed molecular mechanism via TBC1D10B signaling will be critical for understanding the significance of complex signaling crosstalk in the process of FcγR-mediated phagocytosis in macrophages.

# Materials and Methods

### Reagents

DMEM was obtained from Sigma-Aldrich Chemical. FBS was obtained from BioSolutions International. Polyclonal rabbit anti-Rit1 antibody (Sigma-Aldrich), mouse monoclonal anti–glyceraldehyde-3-phosphate dehydrogenase (GAPDH) antibody (AM4300; Ambion), rabbit anti-sheep erythrocyte IgG (ICN55806; Organo Teknika-Cappel) and anti-mouse and anti-rabbit IgG conjugated to HRP (W4021 and W4011; Promega), BSA (Sigma-Aldrich Chemical) were purchased from commercial sources. Rabbit anti-TBC1D10B antibody was generated against a GST-fusion protein containing amino acid residues 1–227 of TBC1D10B. Other reagents were purchased from Wako Pure Chemicals or Nacalai Tesque unless otherwise indicated.

### Cell culture

Mouse macrophage RAW264 and COS-7 cells were obtained from Riken Cell Bank and cultured in DMEM containing 10% heat-inactivated FBS, 100-U/ml penicillin, and 100 µg/ml streptomycin, as described in the manuals of the cell line bank. Before the imaging experiments, the culture medium was replaced with Ringer's buffer (RB) consisting of 155-mM NaCl, 5-mM KCl, 1-mM

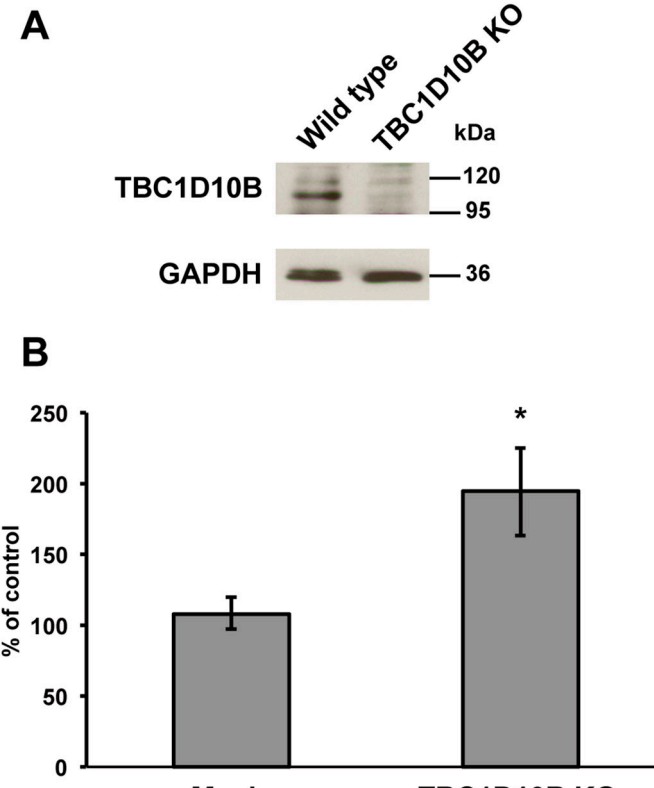

**Figure 8. TBC1D10B-KO promotes phagosome formation.**
**(A)** RAW264 cells were transfected with pSpCas9n(BB)-2A-puro-TBC1D10B encoding Cas9 protein and gRNA for TBC1D10B. Transfected cells were diluted and selected as single clones using puromycin. The depletion of TBC1D10B protein was confirmed by Western blot analysis using an antibody against TBC1D10B. GAPDH protein was used as an internal control. **(B)** Efficiency of IgG-E uptake was calculated based on 50 TBC1D10B-KO cells and 50 cells transfected with pSpCas9n(BB)-2A-puro plasmid (mock) or 50 non-transfected (control) cells. The results are shown as the mean ± SEM percentage compared with control cells for four independent experiments. *$P$ < 0.05, compared with mock-transfected cells ($t$ test).

$MgCl_2$, 2-mM $CaCl_2$, 2-mM $Na_2HPO_4$, 10-mM glucose, 10-mM HEPES-NaOH pH 7.2, and 0.5-mg/ml BSA. BMMs were obtained from the femurs of C57BL/6 mice, as previously described (Araki et al, 1996). The animal experiments were approved by the Animal Care and Use Committee for Kagawa University (approval number: 18602) and were conducted in accordance with the National Institutes of Health Guide for the Care and Use of Laboratory Animals.

### DNA constructs and transfection

To construct pEGFP-Rit1-WT, pTagRFP-Rit1-WT, pEGFP-TBC1D10A-WT, pEGFP-TBC1D10B-WT, pTagRFP-TBC1D10B-WT, and pEGFP-TBC1D10C-WT, cDNA fragments comprising the entire coding region for mouse Rit1, TBC1D10A, TBC1D10B, and TBC1D10C were generated by PCR amplification of mouse cDNAs. The primer sequences for Rit1, TBC1D10A, TBC1D10B and TBC1D10C are as follows: Rit1 forward primer (5′-CAGATCTATGGAGTCCGGAGCTCGCCC-3′) and Rit1 reverse primer (5′-CGTCGACTCAGGTGACCGAGTCTTTCTTC-3′);

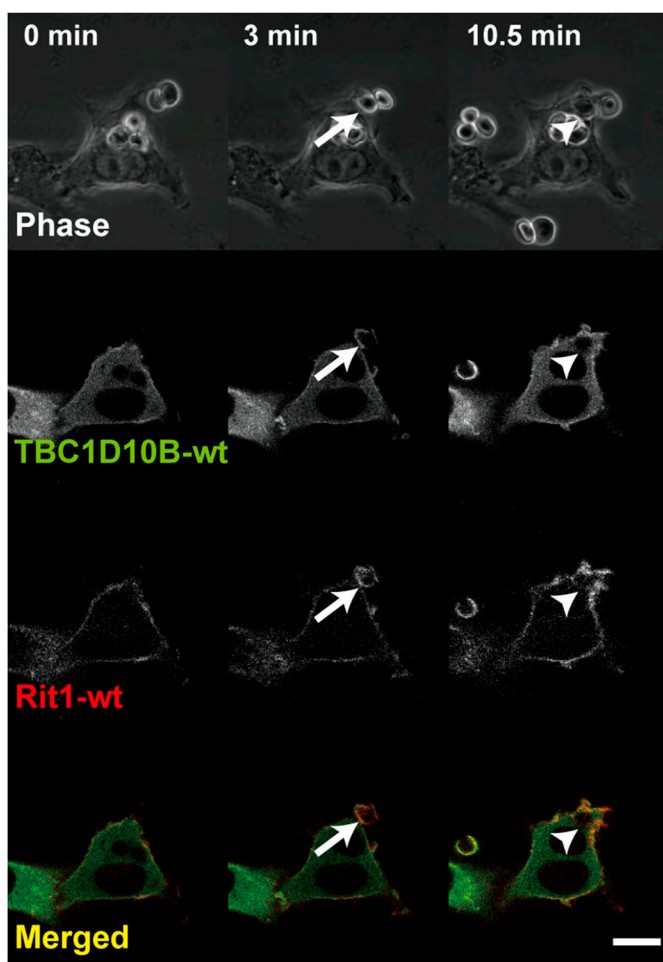

**Figure 9.    Time-lapse imaging showing localization of TBC1D10B and Rit1during FcγR-mediated phagocytosis.**
Time-lapse images of FcγR-mediated phagocytosis in cells co-expressing GFP-TBC1D10B-WT (green) and TagRFP-Rit1-WT (red) were acquired by confocal microscopy. TBC1D10B and Rit1 transiently colocalized at the membrane of phagocytic cups (arrows). Arrowheads indicate newly internalized phagosomes. Representative images are shown. The corresponding video is shown in Video 3. Scale bar: 10 μm.

TBC1D10A forward primer (5′-TCTCGAGCTATGGCGAAAAGCAGCAGAGAG-3′) and TBC1D10A reverse primer (5′-TGGATCCTTACAAGTAGGTGTCCTCACTC-3′); TBC1D10B forward primer (5′-CAGATCTATGGAGACGGGTCCGGCGC-3′) and TBC1D10B reverse primer (5′-TGGATCCTCAAAAGTAAGCATCCTGCCGG-3′); TBC1D10C forward primer (5′-GTCCGGAATGGCCCAGGCCCTGGGAG-3′) and TBC1D10C reverse primer (5′-AGAATTCTTAGAATCGGGTGTCCAGGAAA-3′). The fragments were cloned into pEGFP-C1 (Clontech) and pTagRFP-C (Evrogen). pEGFP-Rit1-Q79L (GTP-locked mutant), pEGFP-Rit1-S35N (GDP-locked mutant), pEGFP-Rit1-WT-W204A (mutant defective in plasma membrane targeting), pEGFP-Rit1-Q79L-W204A (double mutant defective in GTPase activity and plasma membrane targeting), pEGFP-TBC1D10A-R160A (GAP activity–deficient mutant), pEGFP-TBC1D10B-R395A (GAP activity–deficient mutant), and pEGFP-TBC1D10C-R139A (GAP activity–deficient mutant) were generated by means of the QuickChange II site–directed mutagenesis kit (Stratagene). pLifeAct-GFP was generated by substituting GFP

with mTurquoise2 in pLifeAct-mTurquoise2 (plasmid #36201; Addgene). All constructs were verified by sequencing before use. RAW264 cells were transfected using the Neon transfection system (Invitrogen), according to the manufacturer's instructions. The transfected cells were seeded onto 25-mm coverslips and cultured in growth medium. All experiments were performed 24–48 h after transfection.

## RT–PCR

Total RNAs were extracted using a standard extraction protocol (Egami & Araki, 2018). Total RNAs (5 μg aliquots) were converted to cDNAs using Superscript reverse transcriptase (Invitrogen) and oligo (dT) nucleotides. PCR reactions were conducted with specific primers. The primers used were as follows: for Rit1, primers targeted bases 104–123 and 742–763 of GenBank Accession No. BC012694; and for GAPDH, primers targeting bases 178–199 and 537–557 of GenBank Accession No. BC096440. The number of amplification cycles was 33 for the detection of Rit1 and 21 for the detection of GAPDH. PCR products were separated electrophoretically on a 1% agarose gel and visualized by staining with ethidium bromide.

## CRISPR/Cas-mediated knockout of Rit1 and TBC1D10B

pSpCas9(BB)-2A-puro (PX459) and pSpCas9n(BB)-2A-puro (PX462) were gifts from Feng Zhang (plasmids #48139 and #62987, respectively; Addgene) (Ran et al, 2013). For KO of Rit1 in RAW264 macrophages, the following gRNA sequences were tested: 5′-GAGTCCGGAGCTCGCCCCAT-3′ and 5′-GCTACCAATGGGGCGAGCTC-3′. For the knockout of TBC1D10B, 5′-GCTGTAGTCACTGGCGGTCC-3′ and 5′-TTCGGCCCCAGTCGAATTGG-3′ were used. After ligation of the synthesized sequences into pSpCas9(BB)-2A-puro or pSpCas9n(BB)-2A-puro, the pSpCas9(BB)-2A-puro-Rit1 and pSpCas9n(BB)-2A-puro-TBC1D10B plasmids were verified by sequencing. Transfection of plasmids was performed using the Neon Transfection System, as previously described (Egami et al, 2017). At 12 h after transfection, the transfectants were selected with 5 μg/ml puromycin for 24 h. Subsequently, a limiting dilution of the surviving cells was performed. The resultant single colonies were cultured without puromycin. Rit1 and TBC1D10B knockouts were confirmed by both DNA sequencing and Western blot analysis.

## Western blotting

Cells were washed with ice-cold PBS and suspended in lysis buffer containing 50-mM Tris–HCl (pH 7.5), 150-mM NaCl, 1% Triton X-100, 0.5% sodium deoxycholate, 0.1% SDS, 0.1% CHAPS, and protease inhibitor cocktail (Nacalai Tesque). The cell lysates were separated from the pellets after centrifugation at 12,100g for 15 min. Protein concentrations were estimated using the BCA protein assay reagent, and equal amounts of protein were denatured and reduced with a sample buffer containing 1% SDS and 2.5% 2-mercaptoethanol. The samples were subjected to SDS–PAGE and transferred to a polyvinylidene difluoride membrane (Millipore). Western blotting was conducted using the ECL Prime Western Blotting Detection System (Amersham; Cytiva, Danaher Corporation). The

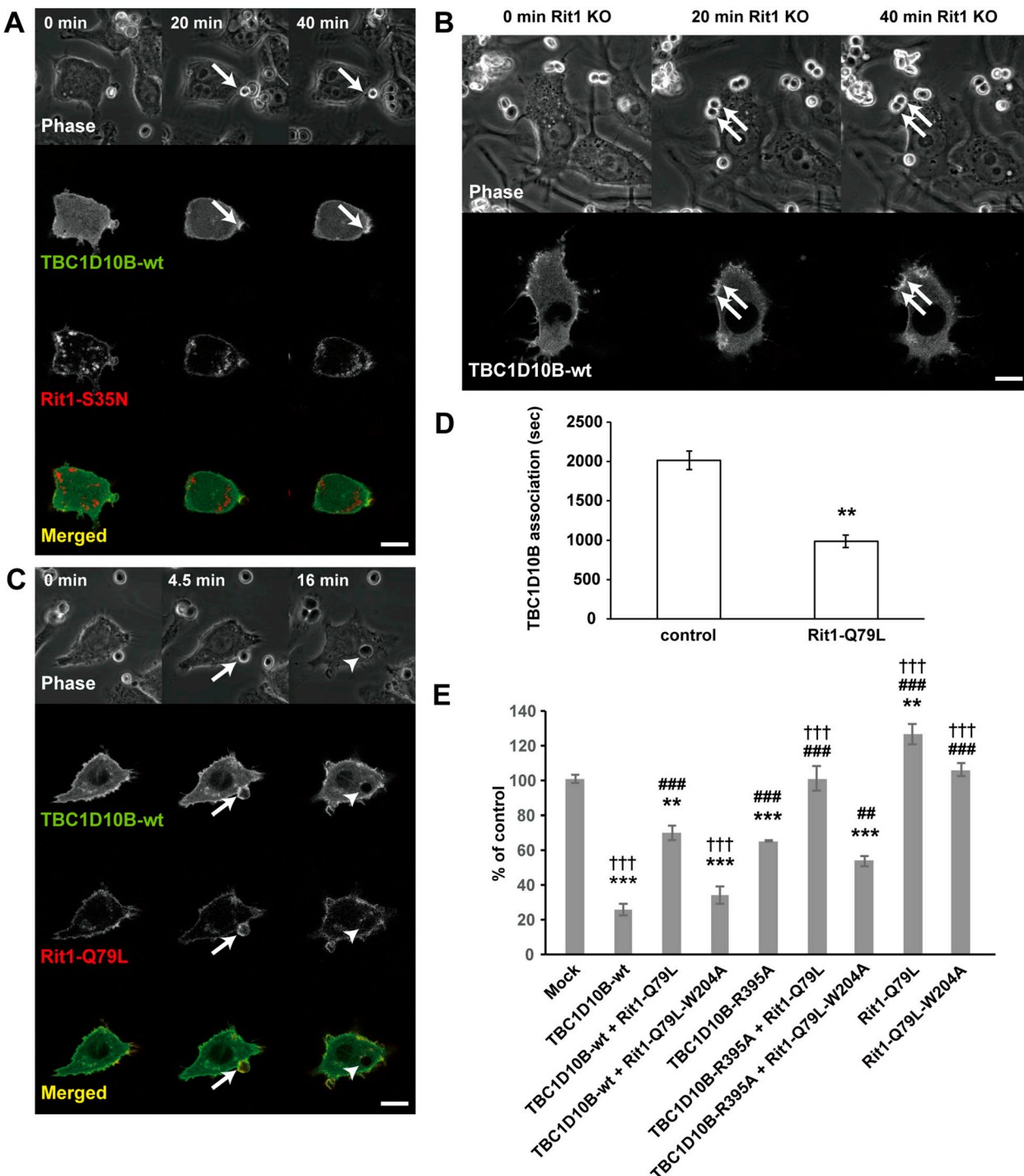

**Figure 10. Expression of the GTP-bound mutant Rit1-Q79L promotes TBC1D10B dissociation from phagocytic cups and restores TBC1D10B-mediated inhibition of phagosome formation.**
**(A, B)** Time-lapse images of cells co-expressing GFP-TBC1D10B-WT (green) and TagRFP-Rit1-S35N (red) (A), and Rit1-KO cells expressing GFP-TBC1D10B-WT (B) were acquired by confocal microscopy. During FcγR-mediated phagocytosis, TBC1D10B remained associated with phagocytic cups for a long time in cells expressing Rit1-S35N or Rit1-KO cells. Scale bar: 10 μm. **(C)** RAW264 cells were co-transfected with GFP-TBC1D10B-WT (green) and TagRFP-Rit1-Q79L (red), allowed to contact with IgG-Es, and observed by confocal microscopy. Scale bar: 10 μm. **(D)** Quantification of the dissociation time of TBC1D10B from the phagosomal membrane during phagosome formation. Live RAW264 cells expressing GFP-TBC1D10B-WT (control) or co-expressing GFP-TBC1D10B-WT and TagRFP-Rit1-Q79L were observed by microscopy. The GFP-TBC1D10B fluorescence intensity at the phagosomal membrane was measured. The fluorescence intensity of GFP-TBC1D10B was normalized by dividing it by the value of the fluorescence intensity of a region in the cytoplasm. We calculated the duration of time (s) during which the normalized values at the phagosomal membrane remained greater than 1. Values represent the mean ± SEM of three independent replicates (n = 3; 20 IgG-Es binding sites in more than five cells in each condition were assessed per replicate). **P < 0.01 compared with control transfected cells (Student's test). **(E)** RAW264 cells transfected with TBC1D10B and/or Rit1 construct(s) were

membrane was blocked with 5% nonfat dried milk in PBS containing 0.1% Tween 20 for 30 min at room temperature and probed with anti-Rit1, anti-TBC1D10B, or anti-GAPDH antibody (1:10,000) at 4°C overnight. After washing, the membrane was incubated with HRP-conjugated anti–rabbit-IgG or anti–mouse-IgG secondary antibody (dilution 1:10,000) for 2 h at room temperature, developed using an ECL Prime reagent, and exposed to Hyperfilm (Amersham; Cytiva, Danaher Corporation).

## Phagocytosis assay

Sheep erythrocytes were opsonized with rabbit anti-sheep erythrocyte IgG and resuspended in PBS, as described previously (Araki et al, 1996). For quantitative analysis of phagosome formation, IgG-opsonized erythrocytes (IgG-Es) were added to RAW264 macrophages grown on coverslips. After 20 min of incubation with IgG-Es at 37°C, the cells on the coverslips were dipped into distilled water for 20 s to disrupt the extracellularly exposed IgG-Es and then fixed with 4% PFA and 0.1% glutaraldehyde. We counted the number of internalized IgG-Es in 50 cells randomly chosen under phase-contrast and fluorescence microscopy. The phagocytic index, that is, the mean number of IgG-Es phagocytosed per cell, was calculated. The index obtained for the transfected cells was divided by the index obtained for the non-transfected (control) cells and expressed as a percentage of the control cells. The data are presented as the mean ± SEM of three or four independent experiments. For the binding assay, RAW264 macrophages were incubated with IgG-Es for 30 min at 4°C, briefly washed in ice-cold PBS to remove unbound IgG-Es, and fixed. The number of cell-bound IgG-Es was counted in 50 cells, and the binding index, that is, the mean number of bound IgG-Es per cell, was calculated. The binding index is presented as a percentage of that in non-transfected (control) cells. The data are presented as the mean ± SEM of three independent experiments. We conducted double-blind counting to confirm our data.

## Live-cell imaging and image analysis

RAW264 cells grown on 25-mm circular coverslips were assembled in an RB-filled chamber on a thermo-controlled stage (Tokai Hit). Phase-contrast and fluorescence images of live cells were sequentially taken using an Axio observer Z1 inverted microscope equipped with a 63×/1.4-NA objective and a laser scanning unit (LSM700, Zeiss) under the control of ZEN2009 software (Zeiss), as previously described (Fujii et al, 2013). Time-lapse images of phase-contrast and fluorescence microscopy were captured at 15-s intervals and converted to MP4 Videos. At least four examples were observed in each experiment. To quantify the recruitment levels of the proteins, we measured the maximum values of GFP fluorescence signal intensity in the phagocytic cup. The value of fluorescence intensity at the phagocytic cup was normalized by dividing

it by the value of the fluorescence intensity of a region in the cytoplasm. We conducted a double-blind image analysis. To assess the dissociation time of TBC1D10B from the phagosomal membrane during phagosome formation, GFP fluorescence signal intensity at the phagosomal membrane was measured in live RAW264 cells expressing GFP-TBC1D10B. The intensity of the GFP signal was normalized to the fluorescence intensity of the cytoplasmic region. We determined the duration of time (s) during which the normalized values at the phagosomal membrane were greater than 1.

## Statistical analysis

Two-tailed $t$ tests or one-way ANOVA followed by Dunnett's or Tukey's test were performed. All $P$-values were considered significant at $P < 0.05$.

## Protein expression and purification

pGEX-4T-3-RGL3, encoding the GST-RalGDS–like 3 (RGL3)-Ras–binding domain, was constructed. We expressed GST–RGL3 or GST–ankyrin repeat (ANKR) proteins in *Escherichia coli* (BL21[DE3]) using methods similar to those described previously (Egami et al, 2015). Cells were grown in LB medium and incubated in the presence of 0.1-mM isopropyl-1-thio-$\beta$-D-galactopyranoside (15 h at 25°C). After centrifugation at 4°C, the resulting cell pellets were resuspended in buffer (20-mM Tris–HCl pH 7.5, 1-mM EDTA, 1-mM dithiothreitol, 0.5-mM phenylmethylsulfonyl fluoride, 50 units/ml aprotinin, 2-μg/ml leupeptin, and 2-μg/ml pepstatin A) and subjected to sonication. Cell debris was removed by centrifugation, and the resultant supernatant was used as an *E. coli* lysate. Purification of the GST-fusion proteins to near homogeneity was achieved using Glutathione Sepharose 4 B affinity chromatography (GE Healthcare). The purity of the samples was at least 80%, as confirmed by Coomassie Brilliant Blue staining of the SDS–PAGE gels.

## GST pull-down assay

COS-7 cells were washed with ice-cold PBS and suspended in lysis buffer containing 25-mM Tris–HCl pH 7.2, 150-mM NaCl, 5-mM MgCl$_2$, 1% NP-40, 5% glycerol, and protease inhibitor cocktail (Nacalai Tesque). The cell lysates were sonicated at 4°C and separated from the pellets after centrifugation at 12,100$g$ for 15 min. Protein concentrations were estimated using the BCA protein assay reagent. Glutathione Sepharose beads coupled to GST–ANKR or GST–RGL3 were incubated for 2 h at 4°C with 500 μg of the cell lysates. After the beads were washed four times with lysis buffer, the proteins bound to the beads were analyzed on 12.5% SDS–PAGE gels, followed by Western blotting. GST–ANKR and GST–RGL3 were stained with Ponceau S.

---

incubated with IgG-Es. The efficiency of phagosome formation was calculated based on 50 transfected and 50 non-transfected (control) cells. The results are expressed as the mean ± SEM percentage comp43ared with control cells for four independent experiments. **$P < 0.01$; ***$P < 0.001$ versus corresponding mock-transfected cells. ##$P < 0.01$; ###$P < 0.001$ versus the corresponding cells transfected with TBC1D10B-WT. †††$P < 0.001$ versus corresponding cells transfected with TBC1D10B-R395A (one-way ANOVA followed by Tukey's test).

# Supplementary Information

# Acknowledgements

The authors would like to thank Dr. Katsuya Miyake for their helpful discussion, and Mr. Kazuhiro Yokoi and Ms. Yukiko Iwabu for their skillful assistance. This study was supported by the Japan Society for the Promotion of Science (JSPS) (16K08468 and 20K07245 to Y Egami) and in part by JSPS (18K06831 and 23K06306 to N Araki and 19K07248 to K Kawai).

## Author Contributions

Y Egami: conceptualization, resources, formal analysis, supervision, funding acquisition, validation, investigation, methodology, project administration, and writing—original draft, review, and editing.
K Kawai: resources and funding acquisition.
N Araki: conceptualization, supervision, and funding acquisition.

## Conflict of Interest Statement

The authors declare that they have no conflict of interest.

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
