## [Reviewer comments · Life Science Alliance]

Life Science Alliance

Rit1-TBC1D10B signaling modulates FcγR-mediated phagosome formation in RAW264 macrophages

Youhei Egami, Kastuhisa Kawai, and Nobukazu Araki

DOI: <https://doi.org/10.26508/lsa.202402651>

Corresponding author(s): Youhei Egami, Kagawa University

Review Timeline:

Submission Date:	2024-02-08
Editorial Decision:	2024-04-15
Revision Received:	2024-07-15
Editorial Decision:	2024-07-18
Revision Received:	2024-07-22
Accepted:	2024-07-22

Transaction Report:

April 15, 2024

Re: Life Science Alliance manuscript #LSA-2024-02651-T

Dr. Youhei Egami
Kagawa University
Department of Histology and Cell Biology, School of Medicine
1750-1 Ikenobe, Miki-cho, Kita-gun, Kagawa Prefecture
Takamatsu 7610793
Japan

Dear Dr. Egami,

Thank you for submitting your manuscript entitled "Rit1-TBC1D10B signaling modulates FcyR-mediated phagosome formation in RAW264 macrophages" to Life Science Alliance. The manuscript was assessed by expert reviewers, whose comments are appended to this letter. We invite you to submit a revised manuscript addressing the Reviewer comments.

Thank you for this interesting contribution to Life Science Alliance. We are looking forward to receiving your revised manuscript.

Sincerely,

B. MANUSCRIPT ORGANIZATION AND FORMATTING:

Reviewer #1 (Comments to the Authors (Required)):

In this manuscript, Egami et al. revealed that the small GTPase Rit1 regulates FcγR-mediated phagocytosis through TBC1D10B. Overall, the manuscript is clear, well written, and the findings are novel. Experiments are well conducted, carefully quantified, and I agree with the interpretation of the data throughout the paper. I don't have any particular concern about the quality of this study. In my opinion, this paper only falls short in terms of overall mechanism, which remains relatively limited. The study convincingly shows that TBC1D10B reduces particle internalization in two ways, dependently and independently of its GAP activity. However, these inhibition mechanisms were not explored. Deciphering the GAP-independent mechanism would be very interesting but could be quite challenging. However, since TBC1D10B is a GAP for Rab35, and a previous study from the same group showed a role of Rab35 in phagocytosis, experiments to determine whether TBC1D10B regulates phagocytosis (at least in part) through Rab35 seems to be a missing element in this study. In addition, it is unclear how Rit1 and TBC1D10B get recruited to the forming phagosome. The authors speculated that Rit1 could be recruited through phosphoinositide. This notion could be easily tested by imaging the recruitment of Rit1 and TBC1D10B in cells treated with PI3 kinase inhibitors. I believe that such experiments would clarify the mechanism by which Rit1-TBC1D10B regulate phagocytosis and would appreciably increase the impact of the study.

Reviewer #2 (Comments to the Authors (Required)):

Egami and collaborators have investigated roles of Rit1 and TBC1D10B in FcγR-mediated phagocytosis in the murine macrophage cell line RAW264.7. They used complementary approaches to demonstrate the early and transient recruitment of these two proteins to the phagocytic cup/nascent phagosome as well as their requirement for FcγR-mediated phagocytosis. Overall, the manuscript is very well written and the data is of high quality and clearly presented, providing a mechanistic aspects of the function of Rit1 and TBC1D10B. The following issues would complete this nice piece of work. .

1- It would be important to demonstrate whether the requirement for Rit1 and TBC1D10B extends to other types of receptor-mediated phagocytosis or is restricted to FcγR-mediated phagocytosis. The authors could assess the internalization of complement-opsonized zymosan and of latex beads. Another issue of interest is whether the requirement for Rit1 and TBC1D10B is related to the size of the phagocytic prey.

2- Given the well-described roles of Rho GTPases in phagocytosis, it would be of interest to determine whether the absence of Rit1 or TBC1D10B affects the recruitment of Rac and Cdc42 to the phagocytic cup during FcγR-mediated phagocytosis.

Reviewer #3 (Comments to the Authors (Required)):

This study addresses the complex regulation of phagocytosis by GTPases, adding to this complexity with new data on a GTPase, Rit1 and a common GAP, TBC1D10B. They reveal a transient association Rit1 with phagosomes during FcγR-mediated uptake of IgG bound erythrocytes, and its functional role is examined, and clearly demonstrated, by a reduction of phagocytosis with GDP-bound Rit1 or more emphatically upon CRISPR knockout of Rit1.

The authors then examine the presence/localisation of TBC-domain containing proteins, as known GAPs for Ras family GTPases. TBC1D10B is dynamically colocalised with Rit1 on phagosomes. TBC1D10B, strikingly reduced phagocytosis of IgG-E and expression of a GAP activity dead mutant still reduced phagocytosis by ~50% suggesting TBC1D10B can inhibit FcγR-mediated phagocytosis, in both Rab-GTP dependent and independent ways. A CRISPR knockout of TBC1D10B increased IgG-E phagocytic activity compared to controls. Further analysis through pull-downs and activation status indicated that TBC1D10B is indeed not a GAP for Rit1. Instead the authors show that Rit1 acts as a membrane recruiter upstream of TBC1D10B, and the two provided rate control for the this uptake process.

This is a pleasing and interesting advance to the field. Overall these are logical and well-crafted experiments and the figures consistently provide clear evidence to back up the conclusions, although sampling in cell labeling and phagocytosis experiments

is arguably limited (see below). The discussion makes the case for Rit1, TBC1D10B as new co-regulators of Ig-mediated phagocytosis, but not in a traditional fashion, with Rab35 and Arf6 discussed as potential members of this scenario; future biochemical or structural studies could explore the overriding complex(es) in further detail.

Minor comments

Nearly all of the figures in the manuscript show a single cell for each condition and quantification is based on manually counting e.g. 50 cells or 30 cups per condition. Phagocytosis can be notoriously variable in fields of cells. The authors need to make some comment about the level of variability between cells in their experiments, or show a low magnification field of cells (for at one of their experiments) or perhaps double-blind their counting to ensure these are representative data.

For the graphs with grey and white bars it would be helpful to have them identified on the figure itself.

Related to Figure 5, did the knockout of Rit1 has any impact on the binding of IgG-E to the RAW264.7 cells? Comment only required.

Referring to Figure 8, It is very striking that the KO of TBC1D10B results in 2 x the level of IgG-E phagocytosis. Presumably, the expression of Rit1-S35N (or indeed Rit1 KO) in these cells would have no impact, unless there is an additional, parallel mechanism by which Rit1 is regulating phagosome formation. Comment only required.

We are grateful to reviewers for the useful suggestions that have helped us to improve our paper.

In the legend for Figure 3, we have removed the words '(green)' and '(red)' because these colors were not used in the figure. Furthermore, we have corrected typos in the manuscript on page 11, line137.

As indicated in the responses that follow, we have taken all these comments and suggestions into account in the new version of our paper. We hope the revised version is now suitable for publication.

Reviewer 1 Comments 1

Since TBC1D10B is a GAP for Rab35, and a previous study from the same group showed a role of Rab35 in phagocytosis, experiments to determine whether TBC1D10B regulates phagocytosis (at least in part) through Rab35 seems to be a missing element in this study.

Response:

In our live-cell imaging analysis, we observed GFP-Rab35-WT colocalizing with TagRFP-TBC1D10B-WT at the membrane of phagocytic cups during FcγR-mediated phagocytosis. Subsequently, we investigated whether overexpression of GFP-TBC1D10B-WT significantly reduced Rab35 activation levels during FcγR-mediated phagocytosis. However, due to the low transfection efficiency of RAW264 cells, we did not detect significant inactivation of Rab35 in cells expressing GFP-TBC1D10B-WT compared to cells expressing GFP (control) in our GST-pulldown assay. Following this, we compared Rab35 activation levels between TBC1D10B KO cells and RAW264 wild-type cells during FcγR-mediated phagocytosis. However, we did not observe a clear difference in Rab35 activation levels between TBC1D10B KO and wild-type RAW264 cells. Given that TBC1D10A, TBC1D10B, and TBC1D10C function as GTPase-activating proteins (GAPs) for Rab35, double or triple knockout (KO) experiments may be necessary to resolve this issue.

Reviewer 1 Comments 2

It is unclear how Rit1 and TBC1D10B get recruited to the forming phagosome. The authors speculated that Rit1 could be recruited through phosphoinositide. This notion could be easily tested by imaging the recruitment of Rit1 and TBC1D10B in cells treated with PI3 kinase inhibitors.

Response:

According to the reviewer's suggestion, we examined the effect of PI3 kinase inhibitors (wortmannin and LY294002) on the localization of Rit1 and TBC1D10B during FcγR-mediated phagocytosis. In RAW264 cells treated with PI3 kinase inhibitor (wortmannin or LY294002), both Rit1 and TBC1D10B accumulated at the membrane of phagocytic cups. Based on the previous reports indicating that Rit1 binds to both PI(4,5)P₂ and PI(3,4,5)P₃ (Heo et al., 2006) and that PI(4,5)P₂ is locally increased in the membrane of pseudopodia (Levin et al., 2015), Rit1 may be recruited to the phagosomal membrane through interaction with PI(4,5)P₂.

Reviewer 2 Comments 1-1

It would be important to demonstrate whether the requirement for Rit1 and TBC1D10B extends to other types of receptor-mediated phagocytosis or is restricted to FcγR-mediated phagocytosis. The authors could assess the internalization of complement-opsonized zymosan and of latex beads.

Response:

We examined the involvement of Rit1 and TBC1D10B in CR3-mediated phagocytosis and the uptake of latex beads. In Rit1 KO cells, the internalization of complement-opsonized zymosan and 4.5-μm latex beads was inhibited. Moreover, the expression of TBC1D10B reduced the uptake of these particles. These data indicate that TBC1D10B and Rit1 are involved not only in FcγR-mediated phagocytosis but also in these other types of phagocytosis.

Reviewer 2 Comments 1-1

Another issue of interest is whether the requirement for Rit1 and TBC1D10B is related to the size of the phagocytic prey.

Response:

We investigated whether the inhibition of Fc γ R-mediated phagocytosis in Rit1 knockout (KO) cells and TBC1D10B-transfected cells was influenced by the size of the phagocytic targets. IgG-opsonized latex beads of 2 μ m and 4.5 μ m were prepared, and phagosome formation was quantified in these cells. In our preliminary experiments, Rit1 knockout or TBC1D10B expression inhibited the uptake of 4.5- μ m latex beads. In contrast, Rit1 KO or TBC1D10B expression did not clearly reduce phagocytosis of 2- μ m particles. A previous report indicated that the phagocytosis of 1–3- μ m particles is relatively unaffected by PI3 kinase inhibitors (Cox et al., 1999). Thus, the inhibition of phagosome formation in Rit1 KO or TBC1D10B-transfected cells might be linked to PI3-kinase signaling.

Reviewer 2 Comments 2

Given the well-described roles of Rho GTPases in phagocytosis, it would be of interest to determine whether the absence of Rit1 or TBC1D10B affects the recruitment of Rac and Cdc42 to the phagocytic cup during Fc γ R-mediated phagocytosis.

Response:

We observed the recruitment of GFP-Rac1 and GFP-Cdc42 to the phagocytic cups during Fc γ R-mediated phagocytosis in Rit1 KO cells and TBC1D10B KO cells. In our live-cell imaging analysis, the depletion of Rit1 had no effect on the accumulation levels of Rac1 and Cdc42 in the membranes of phagocytic cups. Similarly, the recruitment of Rac1 and Cdc42 to the phagocytic cups was not affected by the knockout of TBC1D10B.

Reviewer 3 Comments 1

Nearly all of the figures in the manuscript show a single cell for each condition and quantification is based on manually counting e.g. 50 cells or 30 cups per condition. Phagocytosis can be notoriously variable in fields of cells. The authors need to make some comment about the level of variability between cells in their experiments, or show a low magnification field of cells (for at one of their experiments) or perhaps double-blind their counting to ensure these are representative data.

Response:

We agree with the reviewer's suggestion. We have added the following sentences: "Between 0 and 30 phagosomes per cell were observed" (Figure 2 and 3 legends), "Cells transfected with GFP-TBC1D10A exhibited between 0 and 6 phagosomes per cell. For cells transfected with GFP-TBC1D10B, between 0 and 3 phagosomes per cell were observed. Cells expressing GFP-TBC1D10C showed between 0 and 8 phagosomes per cell" (Figure 6 legend). Furthermore, we conducted double-blind counting of the phagosome formation assay and double-blind image analysis, which yielded similar results. We have included the sentences "We conducted double-blind counting to confirm our data" (page 30, lines 423-424) and "We conducted double-blind image analysis" (page 31, line 437) to the Materials and Methods section.

Reviewer 3 Comments 2

For the graphs with grey and white bars it would be helpful to have them identified on the figure itself.

Response:

We have added bar labels to the graphs in Figure 4 and 7 and revised them to be understandable on their own.

Reviewer 3 Comments 3

Related to Figure 5, did the knockout of Rit1 has any impact on the binding of IgG-E to the RAW264.7 cells? Comment only required.

Response:

The Rit1 knockout had no effect on the binding of opsonized particles to the cells. In the results section, we have added the sentence, "The knockout of Rit1 had no effect on the binding of IgG-E to the cells" (page 10, lines 129-130).

Reviewer 3 Comments 4

Referring to Figure 8, It is very striking that the KO of TBC1D10B results in 2 x the level of IgG-E phagocytosis. Presumably, the expression of Rit1-S35N (or indeed Rit1

KO) in these cells would have no impact, unless there is an additional, parallel mechanism by which Rit1 is regulating phagosome formation. Comment only required.

Response:

We tested the effect of Rit1-S35N expression on the uptake of IgG-Es in TBC1D10B KO cells. In our phagosome formation assay, we found that the expression of Rit1-S35N had no significant impact on phagosome formation.

July 18, 2024

RE: Life Science Alliance Manuscript #LSA-2024-02651-TR

Dr. Youhei Egami
Kagawa University
Department of Histology and Cell Biology, School of Medicine
1750-1 Ikenobe, Miki-cho, Kita-gun, Kagawa Prefecture
Takamatsu 7610793
Japan

Dear Dr. Egami,

Thank you for submitting your revised manuscript entitled "Rit1-TBC1D10B signaling modulates FcyR-mediated phagosome formation in RAW264 macrophages". We would be happy to publish your paper in Life Science Alliance pending final revisions necessary to meet our formatting guidelines.

- please be sure that the authorship listing and order is correct
- please add the Twitter handle of your host institute/organization as well as your own or/and one of the authors in our system
- please use the [10 author names et al.] format in your references (i.e., limit the author names to the first 10)
- we encourage you to revise the figure legend for Figure 1 such that the figure panels are introduced in alphabetical order
- please indicate approval for using mice to obtain femur BMMs

A. FINAL FILES:

B. MANUSCRIPT ORGANIZATION AND FORMATTING:

Sincerely,

July 22, 2024

RE: Life Science Alliance Manuscript #LSA-2024-02651-TRR

Dr. Youhei Egami
Kagawa University
Department of Histology and Cell Biology, School of Medicine
1750-1 Ikenobe, Miki-cho, Kita-gun, Kagawa Prefecture
Takamatsu 7610793
Japan

Dear Dr. Egami,

Thank you for submitting your Research Article entitled "Rit1-TBC1D10B signaling modulates FcyR-mediated phagosome formation in RAW264 macrophages". It is a pleasure to let you know that your manuscript is now accepted for publication in Life Science Alliance. Congratulations on this interesting work.

DISTRIBUTION OF MATERIALS:

Again, congratulations on a very nice paper. I hope you found the review process to be constructive and are pleased with how the manuscript was handled editorially. We look forward to future exciting submissions from your lab.

Sincerely,
